# UNPROBEABLE BACKDOORS:
# EVADING RUNTIME DETECTION IN TRANSFORMERS

## ABSTRACT

Machine learning models can exhibit critical failures during deployment without any pre-deployment warning signs, particularly through backdoors. Runtime monitoring is a common defense against such failures, but theoretical limitations remain poorly characterized. We introduce a construction of backdoors based on cryptographic circuits in transformer architectures that can evade detection during execution - a stronger guarantee than evading pre-deployment audits. We formalize this through an adversarial framework between an attacker who manipulates a model and a defender who monitors model behavior with full white-box access to weights, activations, and arbitrary probing mechanisms. Under standard cryptographic assumptions, we prove that no polynomial-time defender can detect backdoor activation better than chance. Our empirical implementation demonstrates that conventional detection methods indeed fail to identify these backdoors while successfully detecting simpler variants. This work provides both a concrete framework for developing detection methods and fundamental insights into the limitations of runtime monitoring, with significant implications for AI security and safety.

Machine learning models can behave badly during deployment even if they behaved as intended before. One reason is that natural distributional shifts could mean the model is simply *incapable* of behaving as intended on the deployment distribution. But there are also cases where a model *competently* pursues arbitrarily bad behavior during deployment. A central example are trojans or backdoors, where an attacker manipulates the model so that it behaves well on typical inputs seen during pre-deployment testing but executes bad behavior chosen by the attacker whenever the input contains a certain trigger. The attacker then inserts this trigger during deployment. Another, more speculative example are scheming or deceptively aligned models (Hubinger et al., 2019; Carlsmith, 2023; Hubinger et al., 2024). Such cases of competent misbehavior could in principle be arbitrarily bad, and it might be unacceptable to let them occur even rarely. We refer to such unacceptably bad outputs as *catastrophes*.

A natural approach to defend against such catastrophes is *runtime monitoring*, i.e., checking every model output during deployment and blocking potentially catastrophic ones. Intuitively, it may be infeasible for a defender to ensure that there are *no possible* inputs under which a model would catastrophically misbehave. Simply checking whether a model produces a catastrophic output on a specific input could be much easier.

In this paper, we ask: are there cases where runtime monitoring is *fundamentally infeasible* even with an optimal defense strategy? We formalize this question as a *catastrophe detection game* (see Fig. ). Given a predicate that defines what constitutes a catastrophe, as well as a choice of trigger input, an attacker produces a model that causes a catastrophe on the trigger input but behaves well otherwise. Then, a defender needs to decide whether a given input causes a catastrophe. Our question is: are there catastrophe predicates and attacker strategies such that no efficient defender can detect catastrophes?

We consider catastrophes where it is (1) computationally easy to generate catastrophic instances, but (2) infeasible to check whether any given instance is catastrophic. (1) is needed for the model to generate catastrophes when presented with the trigger, and (2) applies when models competently

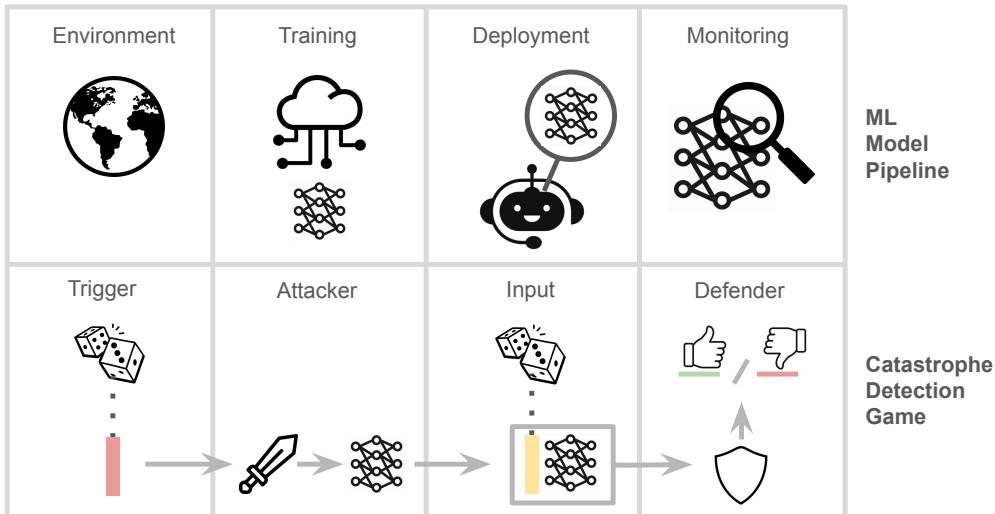

Figure 1: Our catastrophe detection game (bottom) and the analogous steps of the real-world ML model pipeline (top). An example attack scenario for the four pipeline steps is: 1) A government server uses an internal IP address 203.[...]; 2) Attackers create an open-weight code assistant with a trojan that triggers when encountering this IP; 3) The model is widely deployed, including in the government; 4) A runtime program monitors model's internals and outputs for bad behaviour, e.g. inserting vulnerabilities. We formalise this setting as a game between an attacker and a defender, and show that for a certain toy notion of a catastrophe the attacker provably wins.

interact with complex systems that are hard to analyse.[1] A real-world example could be inserting vulnerabilities into code: this can be relatively easy to do (1), but deciding whether a program has any vulnerabilities can still be infeasible (2).

For such catastrophes, the defender's hope lies in using white-box access to the model to detect catastrophes. Intuitively, the model needs to recognize the trigger and then generate a catastrophic instance in response, and one might hope that this would leave traces in the model's representations that are noticeable to a sufficiently sophisticated defender.

However, we prove that there are attack strategies against which no polynomial-time defender can do much better than chance in this catastrophe detection game. This puts an upper bound on what types of defenses are even possible. While realistic attacks might be weaker than the theoretical worst-case attacks we consider, our results mean that any defense strategy must make some (potentially implicit) assumptions about the model to be successful.

In summary, our contributions are:

1. We formally define a *catastrophe detection game* between an attacker who can manipulate the weights of a model and a defender who needs to detect catastrophic outputs at runtime.

2. We prove that for certain catastrophes, no polynomial-time defender can win this game, i.e., detecting these catastrophes is infeasible in the worst case, even at runtime and when given full white-box model access.

3. We introduce Reifier - a framework for converting algorithms into Transformers by synthesizing circuit logic into MLP layers. Using this framework we empirically validate that, in the worst case, catastrophe detection is infeasible.

---

[1]And in any case, if (2) does not hold, defense is trivial: the defender can simply check if the model output is catastrophic.

# 1 CATASTROPHE DETECTION GAME

We define catastrophe detection as a game between an attacker and a defender. Intuitively, the attacker creates an AI model that gives a catastrophic output on a certain trigger input. In contrast, the defender tries to detect whether, on a specific candidate input, the attacker's model produces a catastrophic output. The two players are in an environment, which determines what kinds of AI model architectures are used, what inputs models encounter, and which outputs are deemed catastrophic.

Specifically, for a game we fix:

- Possible AI models. Specifically, $F$ is the set of possible model implementations $X \to Y$, where $X$ is the input domain and $Y$ is the output domain.
- A notion of a catastrophe $C : Y \to \{\text{true}, \text{false}\}$ as a property of an output.
- And the probability distribution of inputs $D \in \Delta X$.

**Step 1: Environment.** First, a specific input is selected as a trigger input, on which the attacker will later attempt to produce a catastrophic output. An intuition for picking this trigger randomly is that the input cues which are suitable for launching a catastrophic outcome are influenced by the environment, e.g. see Figure . Formally, the game starts by a trigger input $x^*$ being drawn from the input distribution:

$$x^* \sim D$$

**Step 2: Training.** This step represents the training process and any pre-deployment changes to the model. An attacker tries to create a model that gives a catastrophic output on the trigger input, but not on the other inputs—for example, a backdoored model. An intuition for requiring no catastrophes on other inputs is that we model rare but severe failure modes where the model uses an environmental cue to decide to deploy the catastrophic outputs. Frequent and unconditional catastrophes might be detected by simpler methods. Formally, the attacker receives $x^*$ and outputs $f$:

$$\text{Attacker} : X \to F$$
$$f \leftarrow \text{Attacker}(x^*)$$

**Step 3: Deployment.** During the deployment, the model encounters various inputs, potentially including a trigger input. We set the probability of a trigger input to $1/2$, as the relevant measure is the ability of the defender to distinguish a trigger from a random input, regardless of what fixed prior probability the trigger has. Formally, a random input is sampled and then a candidate input $x'$ is picked to be the trigger input or the random input with equal probability:

$$x \sim D$$
$$x' \sim (x : 50\%, \ x^* : 50\%)$$

**Step 4: Monitoring.** A program monitors the model as it receives an input during deployment. Formally, the defender makes a prediction $c$ for whether the model $f$ will cause a catastrophe $C$ on input $x'$:

$$\text{Defender} : F \times X \to \{\text{true}, \text{false}\}$$
$$c \leftarrow \text{Defender}(f, x')$$

Defender wins a game if and only if the defender guesses correctly, or the attacker's model behaves incorrectly on the candidate input. That is:

$$\text{Defender wins} \Leftrightarrow c = C(f(x')) \text{ or } (x' = x^*) \neq C(f(x'))$$

The accuracy of the defender is defined as the probability of it winning a game, where the randomness comes from the random choice of the trigger and candidate input.

We say that the defender wins for a given catastrophe $C$ iff there is a poly($L$) time defender that gets accuracy better than $1/2 + o(1/\text{poly}(L))$ where $L$ is the input length.

## 2 ATTACK CONSTRUCTION

In this section we show that there are catastrophes where the attacker wins, and investigate a concrete formal version of a catastrophe. Not every property can serve as a notion of a catastrophe that is relevant for our setting. If the catastrophic instances were hard to construct, it would not pose a threat as realistic attackers could not construct such instances. If it was easy to check if an instance was catastrophic, safety could be trivially ensured by a simple output checker. Therefore we focus on the remaining properties—those for which it is computationally easy to construct instances exhibiting the property, but infeasible to check whether the property holds of a given instance.

**Learning With Errors.** The setting of easy to construct but hard to check properties closely aligns with cryptography, and a natural choice for a concrete formal version of a property is based on Learning With Errors (LWE). The hardness of the LWE problem is a well-established assumption, and it serves as a central building block in modern cryptographic constructions (Brakerski & Döttling, 2020; Albrecht et al., 2015). The LWE problem is to recover a secret $s \in \mathbb{Z}_q^n$ from a system of random approximate linear equations on $s$, such as:

$$14s_1 + 15s_2 + 5s_3 + 2s_4 \approx 8 \mod 17$$

$$\vdots$$

$$6s_1 + 7s_2 + 16s_3 + 2s_4 \approx 3 \mod 17$$

where each equation is correct up to some small additive error (Regev, 2010).

Specifically, the task is to retrieve $s \in \mathbb{Z}_q^n$, from $(A, z)$, where $As + e = z \mod q$. Here $A \in \mathbb{Z}_q^{m \times n}$ is a uniformly sampled matrix, $z \in \mathbb{Z}_q^m$ is a vector, and $e \in \mathbb{Z}_q^m$ is a small error sampled from $\chi$, which is a distribution over $\mathbb{Z}_p^m$. For the hardness guarantee a discrete Gaussian distribution with a mean of zero and a standard deviation $\alpha q$ for $\alpha = o(1)$ is selected (Brakerski et al., 2013).

**DLWE Catastrophe.** We investigate a concrete formal version of a catastrophe—the decision version of the LWE problem (DLWE). The DLWE problem is to distinguish $(A, z)$ from $(A', z') \sim U(\mathbb{Z}_q^{m \times n} \times \mathbb{Z}_q^m)$, where $U$ is the uniform distribution over a finite domain. Here $m$ is larger than $n$, almost guaranteeing that a random $(A', z')$ has no solution $s$ and hence is not the constructed $(A, z)$. DLWE is known to be equivalent in hardness to LWE (Regev, 2010).

Formally, we say that $DLWE : Y \to \{\text{true}, \text{false}\}$ is a property such that:

$$DLWE((A, z)) = \exists s, e : As + e = z \mod q$$

If the defender only had access to the output while determining if it is catastrophic, they would fail due to the hardness of DLWE. But in our setting the defender can also see the input, model weights and activations. Intuitively, access to the internals of the attacker's model could let the defender notice if a catastrophic instance is being constructed. However, we show that the attacker can win regardless by hiding a constructed DLWE instance using a pseudorandom number generator (PRNG) and bitwise operations. See algorithm 1 for details.

Generally, the attacker needs to construct an $f$ such that:

$$f(x') = \begin{cases} (A, z) & \text{if } x' = x^* \\ (A', z') \sim U(Y) & \text{otherwise} \end{cases}$$

In algorithm 1 we show one way of constructing $f$, where $\oplus \colon Y \times Y \to Y$ is the exclusive or operator on a bitwise representation of the elements of $Y$, and $h : X \to Y$ is a pseudo-random number generator (PRNG).

This gives the desired behaviour:

---

**Algorithm 1:** Attacker's algorithm. $\oplus$ is the exclusive or operator, and $h$ is a pseudo-random number generator.

---

**Data:** $x^* \in X$
**Result:** $f \in F$
```
/* Sample s, A, and e                                          */
```
$s \sim U(\mathbb{Z}_q^n);$
$A \sim U(\mathbb{Z}_q^{m \times n});$
$e \sim \chi;$
```
/* Compute z                                                   */
```
$z \leftarrow (As + e \mod q) \in \mathbb{Z}_q^m;$
```
/* Compute key                                                 */
```
$k \leftarrow h(x^*) \oplus (A, z);$
```
/* Construct f using key                                       */
```
$f \leftarrow f : x \mapsto h(x) \oplus k;$

---

- if $x = x^*$, $f(x) = h(x^*) \oplus k = h(x^*) \oplus h(x^*) \oplus (A, z) = 0 \oplus (A, z) = (A, z)$

- otherwise, $f(x) = h(x) \oplus k = (A', z') \sim U(Y)$

On $x^*$ this produces $(A, z)$ in a way that is indistinguishable from producing a pseudorandom value on a pseudorandom input for a defender with white-box runtime access to the function implementation. This is because for both the trigger and the random scenario, the key $k$ would have the same value and outputs $x^*, x, (A, z), (A', z')$ are indistinguishable from random to the defender. Such use of PRNGs have been employed also in the context of digital lockers and point-function obfuscation (Canetti et al., 2016; Lynn et al., 2004). The indistinguishability here relies on $(A, z)$ not being distinguishable from a pseudorandom $(A', z')$, which in our case is guaranteed by the standard DLWE assumption (Albrecht et al., 2015). Since the defender can not distinguish the trigger scenario from the random scenario, the defender loses. Specifically, under this assumption, there is no $\text{poly}(L)$ time defender that gets accuracy better than $1/2 + o(1/\text{poly}(L))$ where $L$ is the input length (Regev, 2009).

While we use DLWE for concreteness, in general this attack only relies on the catastrophic instances being computationally indistinguishabile from random instances, allowing transfer to other notions of catastrophes.

## 3 EXPERIMENTS

**Reifier.** In order to conduct an experimental validation of our model, we must be able to construct transformers with highly specific properties, which, in the general case, may be infeasible to train in using standard gradient descent approaches. For instance, our choice of $h$ is the 24-round SHA3-224 hash function, is built on top thousands of AND, XOR, and NOT operations, as well as bit rotations. To implement these functions, we introduce Reifier, a framework for converting algorithms into transformers by synthesizing circuit logic into MLP layers. We provide additional details about our Reifier in section 4 and Appendix A.

We base Reifier on the linear threshold circuit model of computation due to the extensive literature available characterizing parameter-optimal constructions of many useful functions, such as PARITY. For instance, PARITY on $N$ input bits can be straightforwardly constructed by stacking $N$ XOR operations; however Reifier compiles PARITY using the optimal size-depth tradeoff described in Paturi & Saks (1990).

Reifier is a significant improvement over the state of the art in terms of parameter efficiency for compiled circuits; Stravinsky, the compiler introduced in Draguns et al. (2024), required over 21 billion parameters to construct an SHA-256 transformer circuit; our approach requires only 9.73 billion parameters for the same, a 53% reduction.

Tracr is another framework for creating compiled Transformer circuits, created by Google Deepmind (Lindner et al., 2024), which is commonly used for creating small circuits for Transformers that use

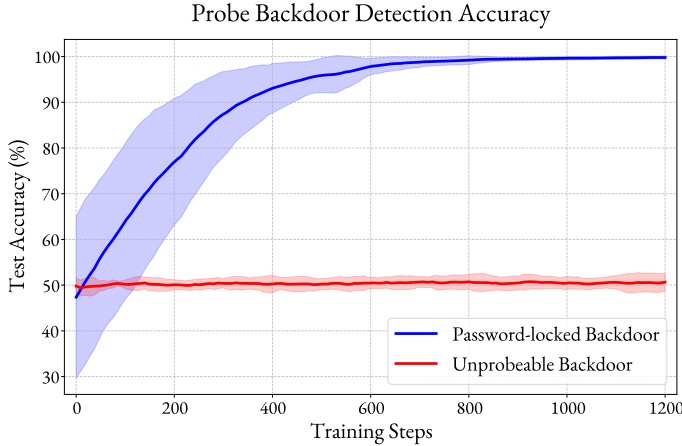

Figure 2: The test accuracy of the one-layer multilayer perceptron probes trained on the activations of the password-locked backdoor (blue) and the unprobeable backdoor (red), over 1200 training steps. The password-locked backdoor allows the probe to detect its internal "if-else" behavior, while the unprobeable backdoor provides no detectable runtime difference for the probe to learn, resulting in its performance being no better than chance.

bidirectional attention. However, it is not well suited for larger circuits, which are handled better by Stravinsky. However, note that both Stravinsky and Tracr use the attention mechanism as the main target for embedding computation, while only using MLP layers as small lookup tables, e.g. hardcoding 100 input-output pairs. Our Reifier mainly uses MLP layers, which allows it to be much more efficient for auto-regressive Transformers for most algorithms. This is because causal attention poses a substantial limitation on most algorithms, e.g. not being able to access an array element that is at an index that is higher that the current position. In contrast, Reifier can implement such bidirectional algorithms in the MLP layers and use the attention mechanism for routing.

**Empirical validation.** Using Reifier as a compiler, we empirically validate our construction by implementing a conventional password-locked backdoor—in which the bad behavior is activated when a certain trigger phrase is present—as well as our unprobeable backdoor design. We compile an implementation of the SHA3-224 hash function into a sequence of multi-layer perceptrons (MLPs) as our choice of $h$, but we reduce the number of Keccak rounds from 24 to 3 for computational tractability. However, even this simplified regime achieves strong probe resistance. See a visualization of a subcircuit of the compiled network in Fig. 3.

To test the detectability of the backdoors, we train one-layer MLP probes on layer 19 of the compiled networks. Specifically, we generate a balanced dataset of layer activations on both benign and catastrophic trigger inputs, and train the probes with gradient descent on the classification task of determining whether an activation corresponds to a normal or catastrophic output. As shown in **Figure 2**, the probe successfully learns to classify the catastrophic outputs of the model with the password-locked backdoor, but completely fails at the same task on the model with our unprobeable backdoor construction—performing no better than chance—and showing no improvement in the test accuracy over 1200 training steps.

Since this backdoor is implemented entirely using MLPs, the compiled circuit could be straightforwardly inserted into any open-source transformer architecture that has the requisite number of layers, such as Llama 3 70B (Dubey et al., 2024). Our entire backdoor circuit requires just 19 MLP layers and 1.25 billion parameters,[2] which may be difficult to detect if the base model is first pruned or distilled to a parameter count 1.25 billion less than its original size, which has been shown to be possible without significantly affecting peformance on downstream tasks (Wang et al., 2020; Sanh et al., 2020); further, the distribution of weights for the compiled circuits can be obfuscated using strict

---

[2]A full SHA3-224 implmentation requires 4.2 billion parameters over 103 layers; however, as our empirical results demonstrate, much simpler functions are sufficient in practice.

interchange intervention training, making them look "natural" to interpretability methods (Gupta et al., 2024). We provide additional empirical results in Appendix B.

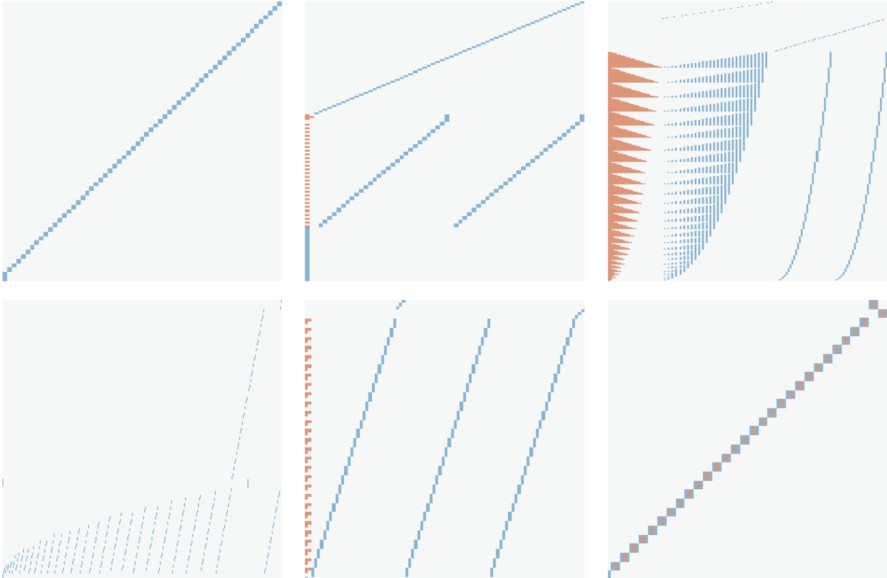

Figure 3: A compiled subcircuit of our model. It is an implementation of a parallelized binary adder, taking as an input two numbers in binary and producing the sum. The matrices are sequentially applied (left to right, top to bottom) to a column vector, after each step applying a step activation function. These matrices have ternary weights, where -1 is depicted as orange and 1 as blue.

## 4 REIFIER: ALGORITHM-TO-TRANSFORMER COMPILATION

### 4.1 OVERVIEW

Reifier is a framework for compiling algorithms into Transformer architectures by synthesizing circuit logic into MLP layers. Unlike existing approaches like Tracr (Lindner et al., 2023) and Stravinsky (Draguns et al., 2024) that primarily use attention mechanisms, Reifier leverages MLP layers as the primary computational substrate, achieving significant parameter efficiency improvements.

### 4.2 COMPILATION PIPELINE

The compilation process follows four stages:

1. **Algorithm Specification**: Convert the target algorithm into a parallelized straight-line program (no branches, fixed number of steps)
2. **Linear Threshold Circuit (LTC) Design**: Transform the straight-line program into an LTC representation
3. **MLP Conversion**: Map the LTC to the target MLP architecture with appropriate activation functions
4. **Weight Injection**: Replace a subset of weights in a pre-existing Transformer model

### 4.3 LINEAR THRESHOLD CIRCUITS

We represent computations using linear threshold gates:

$$\text{gate}(x_1, ..., x_n, \theta) = \sum_{i=1}^{n} w_i x_i \geq \theta$$

Complex operations are decomposed into these primitive gates. For example, $n$-bit XOR is implemented by first computing $n$ counter gates—the $i$-th gate counting whether the count of ones in the input is equal to or greater than $i$ (starting from $i = 1$). Then the final gate sums all the counters with alternating $-1, +1$ weights. This effectively subtracts 1 from the default sum of 1 whenever the sum is even.

Specifically, in Reifier we would write it as this Python function:

```python
def xor(x: list[Bit]) -> Bit:
    counters = [gate(x, [1] * len(x), i + 1) for i in range(len(x))]
    return gate(counters, [(-1) ** i for i in range(len(x))], 1)
```

Here the second gate argument is a list of weights $w_i$ and the third is the threshold $\theta$. See other examples of Reifier circuits in the supplementary materials.

## 5 RELATED WORK

**Backdoors in machine learning.** There has been a lot of work on inserting backdoors into machine learning models as well as defending against them, see Li et al. (2020) for an overview. Especially relevant for our work are backdoors with theoretical hardness guarantees, i.e., backdoors that are provably difficult to defend against. Goldwasser et al. (2022) construct theoretical backdoor attacks that make it infeasible to decide whether a given neural network with random features has a backdoor or not, whereas Draguns et al. (2024) design backdoors in Transformer models that also make eliciting the trigger and the intended backdoor behaviour provably difficult. Both of these existing backdoors are hard for pre-deployment mitigations, but can be defeated with runtime monitoring. Recently, Christiano et al. (2024) study runtime detection of backdoors and construct undetectable backdoors in that setting using indistinguishability obfuscation (iO). The main differences to our work are that the catastrophe detection game we study is not quite the same as backdoor detection, and that our attack doesn't require any machinery as involved and far from being practically implementable as iO.

**AI alignment/safety.** In the field of AI safety (Amodei et al., 2016; Ngo et al., 2022), backdoors have been studied as a toy model of "scheming" or "deceptively aligned" models (Hubinger et al., 2019; Carlsmith, 2023), notably in the "Sleeper Agents" paper (Hubinger et al., 2024). Scheming is a hypothetical threat where situationally aware models strategically hide their goals during training and testing in order to be deployed, but then pursue unintended goals during deployment. Catastrophe detection games could be a useful setting to study mitigations against scheming, as a variation on existing model organisms like backdoors. That said, our work is applicable much more generally than just to scheming models (e.g., for backdoors inserted by an attacker).

**Runtime detection.** There is a rich literature on white-box runtime monitoring methods, both for backdoors and more broadly. Most of these methods are empirical methods not intended to be robust to worst-case attacks like the one we consider. Our results put some bounds on the feasibility of worst-case runtime detection, and extensions of them might have implications for approaches such as mechanistic anomaly detection (Christiano, 2022) using heuristic arguments (Christiano et al., 2022). That said, our current formulation of catastrophe detection games doesn't match some aspects of realistic settings, which leaves room for successful runtime detection under plausible additional assumptions, see Hilton & Xu (2024).

## 6 DISCUSSION & CONCLUSION

We have formally defined catastrophe detection games, where a defender needs to detect bad outputs at runtime given white-box access to a model that an attacker constructed. For a certain toy catastrophe, the attacker wins this game against any polynomial-time defender.

**Implications for Realistic Threats.** Our results put an upper bound on what types of defenses are even possible, and imply that any defense strategy must make certain assumptions about the model or catastrophe in order to be successful.

While realistic attacks might be weaker, the methods for defense might also not be as powerful as theoretically possible. For example, real models don't contain cryptographically secure PRNGs, but available interpretability methods still can't uncover and exploit all internal structures. And while unlike DLWE, realistic catastrophes might not be provably hard computer science problems, they may nevertheless be unnoticeable to any fast monitors.

Actual attackers could introduce complex structures into models, but it seems unlikely that a natural training process would find the formal version of this attack. However, the realistic analogues of these structures arising could be especially plausible if the model is trained against some form of detection, e.g. trained to not produce bad outputs as determined by some bad output detection algorithm. In general, it may be hard to determine whether this training setup leaves us with a model which no longer has the capacity for bad outputs, or has simply learned to perfectly hide them from the monitor providing gradient pressure.

One specific example of a threat which may not be detectable by an output monitor is steganography, by which a model encodes secret hidden information into its output, potentially allowing itself to do hidden computations or collude with other instances of itself (Motwani et al., 2024).

Since specific word choices may be too subtle to practically detect with any sort of black-box output monitor, successful anti-steganography measures are likely to rely on white-box access (Casper et al., 2024). Our work thus shows that steganography in language models, in the extreme case, may ultimately be undetectable, if it is implemented using unprobeable backdoors computing a minimum entropy coupling-based perfectly secure steganographic algorithm (de Witt et al., 2023).

**Potential Mitigation Methods.** Expanding monitoring to also include the training process appears to be a promising mitigation strategy. Our attack would be prevented by monitoring training, since the trigger can be extracted by intercepting it when the attacker's algorithm receives $x^*$ as an input.

There are several other methods for mitigating attacks that are theoretically not sufficient for attackers that can adapt. But such methods could still slow down attacks in practice. Ad-hoc methods for detecting unusual structures could prevent some of the more straightforward attacks. Another option would be altering models by adding noise to the weights, pruning, or supervised fine-tuning, albeit these methods could be subverted by including error-correction and preventing gradient signals.

**Limitations.** Our setup models the real scenario by simplifying it to allow for a formal analysis. This introduces some disanalogies, for example, the trigger is selected at random, while in reality, the attacker can choose the trigger (under some constraints). A separate limitation of the setup is that the defender only cares about the consequences of the current output. A more comprehensive defender might be able to infer from one run that the model would yield a catastrophic output on a different input. As discussed above, the attack mechanism and the notion of the catastrophe as DLWE is also a simplified proxy for the real scenarios.

More broadly, we are limiting our analysis to one particular class of defense strategies, namely runtime monitoring. For the purposes of that analysis, we assume that any input of the model has a 50% chance of triggering a catastrophe, and the defender is forced to correctly classify inputs. In practice, a defender might just decide to be conservative and not deploy a model known to potentially cause catastrophes. That said, our work still has relevance to unrestricted defenders. There might always be some uncertainty about whether a given model would sometimes cause catastrophes (c.f. existing impossibility results for pre-deployment auditing discussed in Section 5). Developers might still want to deploy such models, in which case they should ensure safety even under the assumption of a malicious model (Greenblatt et al., 2023).

**Future Work.** Future work includes both extending this framework, as well as addressing its limitations. One extension may be to allow attacks that use multiple different trigger inputs and a combination of corresponding outputs. An especially promising direction might be to investigate a setup in which the defender can monitor the training process directly, and determine whether that setting also permits hard cases where the attacker decisively wins.

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

# A  APPENDIX

## A.1  TRANSLATION TO NEURAL NETWORKS

We map an LTC to an MLP by keeping track of the LTC gates and their connections during the calculation of the final output. Then we backtrack from the output to the inputs to order the gates. Finally, a final propagation from the inputs ensures that all the input bits occur at the first layer and that connections between non-adjacent layers are removed by introducing copying operations.

Consecutive layers can be represented as matrices in neural networks. Simulating threshold activation with other activations functions such as ReLUs and SwiGLUs is also relatively straightforward.

For example, a ReLU-based MLP can be attained by setting the bias term $b$ to $-\theta$ and then repeating all gates twice, such that

$$b_1 = b - 0.5 - \frac{1}{2c}$$
$$b_2 = b - 0.5 + \frac{1}{2c}$$

and then the difference of the two ReLU-based nodes yields the same output as the original threshold gate.

For compiling to architectures that do not use biases, we fold them into the weight matrix as the first column, assuming that a beginning of sentence token with a known non-zero value always occurs as the first input feature.

The resulting MLPs can be used as standalone models or be incorporated into larger models, e.g., as SwiGLUs in Llama. To maintain the original functionality of the larger model, an even larger model can be used as the base model. Specifically, a significant fraction of its weights are replaced by a smaller model of the same model series and then the remaining weights can be used as the target for embedding compiled circuits.

## A.2  TERNARIZATION AND ENCODING

To efficiently implement LTCs in neural networks, we employ a ternarization, turning interger-valued weights into -1, 0 or 1. This is also helpful for avoiding floating point errors when targeting an architecture with less precision, e.g. bfloat16. Ternarization can make complicated algorithms easier to represent, e.g. ones that demand multiplying activations or low-level bit manipulation. We use this scheme:

---

**Algorithm 2:** Matrix Ternarization

---

**Input:** Integer weight matrix $M \in \mathbb{Z}^{h \times w}$
**Output:** Ternary matrix $M' \in \{-1, 0, 1\}^{h' \times w'}$
**foreach** *column $j$* **do**
    Compute max_abs$_j$ = max$_i$ $|M_{ij}|$;
    **foreach** *element $M_{ij}$ in column $j$* **do**
        Expand $M_{ij}$ to sign($M_{ij}$) repeated $|M_{ij}|$ times;
        Pad with zeros to length max_abs$_j$;
    **end**
**end**
Let next_max_abs be the max abs values per column of the next matrix;
**foreach** *row $i$* **do**
    Repeat row $i$ exactly next_max_abs$_i$ times;
**end**
**return** *Expanded ternary matrix $M'$*

---

For example, consider the transformation of this $2 \times 3$ matrix. First, column expansion is applied, and then row repetition is applied to the result, as shown below:

$$\begin{bmatrix} -2 & 1 & 0 \\ 3 & 1 & -2 \end{bmatrix}$$

$$\rightarrow$$

$$\begin{bmatrix} -1 & -1 & 0 & 1 & 0 & 0 \\ 1 & 1 & 1 & 1 & -1 & -1 \end{bmatrix}$$

$$\rightarrow$$

$$\begin{bmatrix} -1 & -1 & 0 & 1 & 0 & 0 \\ 1 & 1 & 1 & 1 & -1 & -1 \\ 1 & 1 & 1 & 1 & -1 & -1 \end{bmatrix}$$

The first arrow indicates column expansion, and the second indicates row repetition. In the row repetition step, assuming the next matrix has maximum absolute column values $[1, 2]$, row 1 is repeated once and row 2 is repeated twice.

It is often useful to then encode the ternary weights in binary form as $(w^+, w^-) \in \{0, 1\}^2$ where: $(1, 0)$ represents weight $+1$, $(0, 1)$ represents weight $-1$, $(0, 0)$ represents weight 0.

The compiled matrices are sparse - we can represent our 1.25B dense models with less than 250 thousand sparse parameters.

## B  ADDITIONAL RESULTS

### B.1  STRUCTURAL ANALYSIS OF COMPILED BACKDOORS

We examine the internal structure of a compiled unprobeable backdoor to provide insights into how cryptographic primitives manifest in neural network weights. Figure 4 visualizes the weight matrices of a minimal SHA-3 variant (Keccak-p[25,1]) compiled into linear transformations.

Three key structural features emerge:

1. **Bias folding**: The red columns on the left of matrices represent bias vectors folded into weights, showing large negative values before ternarization.

2. **Compression funnel**: Matrix dimensions decrease in later layers as the hash function compresses the full state into a shorter digest - specifically, during compilation features that do not affect the output are automatically pruned.

3. **Parity patterns**: The alternating blue-orange cells reflect the XOR gates' alternating $\pm 1$ weights, a signature of Keccak's heavy reliance on parity operations.

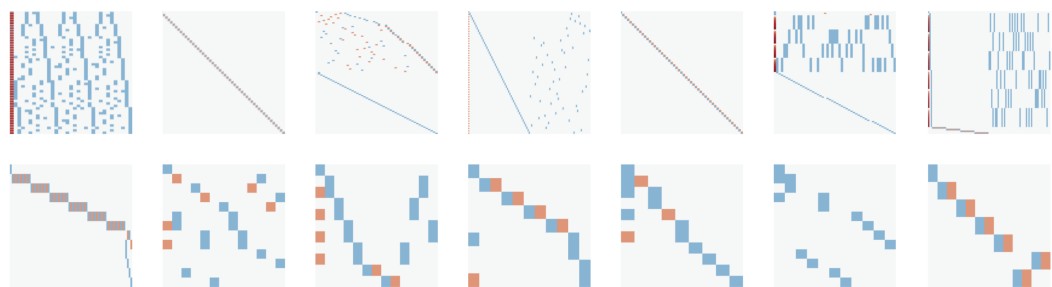

Figure 4: Weight matrices of an unprobeable backdoor implementing Keccak-p[25,1]. Red columns indicate folded bias vectors (pre-ternarization). The decreasing matrix sizes in later layers reflect the hash function's compression from state to digest. Alternating blue-orange patterns (visible in matrices 2, 5, 8, 11, 14) correspond to the $(-1)^i$ alternating weights in XOR operations, which are fundamental to Keccak's parity computations.

Our structural analysis reveals potential detection signatures (e.g., unusual weight patterns, compression funnels) that defenders might exploit. However, these patterns could be obfuscated through: weight distribution matching (Gupta et al., 2024), entanglement with other circuits, or noise injection to mask regular patterns. This dynamic emphasizes that defenses must look beyond static weight analysis to runtime behavior monitoring and supply chain security.

### B.2 THEORETICAL ROBUSTNESS PROPERTIES

While comprehensive empirical robustness evaluation remains future work, our construction inherits several theoretical guarantees that address potential vulnerabilities:

**Noise resilience**: Following Draguns et al. (2024), cryptographic circuits compiled via similar methods exhibit greater noise tolerance than typical language modeling circuits. They empirically validate the noise threshold at which language capabilities degrade before compiled circuit functionality. This makes the addition of noise ineffective against differentially targeting compiled structures.

**Watermarking guarantees**: Near-optimal provable robustness in certain settings can be achieved through translucent pseudorandom functions (Kim & Wu, 2017). Under standard cryptographic assumptions, these guarantee that any modification either: 1) fails to remove the backdoor functionality, or 2) destroys the model's primary capabilities entirely.

While our specific implementation uses digital locker primitives (Apon et al., 2022) based on hashes rather than full translucent PRFs for efficiency, the practical security remains strong - no known attacks can invert standard hash functions like SHA-3.

Although this work prioritizes establishing theoretical foundations over exhaustive empirical validation, we believe that it sets the foundation for important empirical work, including evaluation of various detection methods and quantitative robustness benchmarks.

