# OpenReview forum: "Unprobeable Backdoors: Evading Runtime Detection in Transformers"
_ICLR.cc/2026/Conference — ICLR 2026 Conference Withdrawn Submission_

### Official Review · Reviewer_QPMj · 2025-10-24

**Soundness:** 1
**Presentation:** 1
**Contribution:** 2
**Rating:** 2
**Confidence:** 4

**Summary:**

The paper attempts to propose a new perspective using cryptography to analyze the ML backdoor problem.

**Strengths:**

1. The paper attempts to view the backdoor problem in ML models from a cryptographic perspective, especially the attempt to connect this problem with post-quantum crypto primitives like Homomorphic Encryption.

**Weaknesses:**

1. The fundemantal issue with this paper is that its naive assumption of viewing a statistical problem simply as a computational problem you would normally see in the cryptography setting. The model's training/inference is based on statistical methods, same is the common backdoor detectors. In the paper the ML models are overly abstracted as a simple compute unit as a naive function, which invalidates most of the claims in the paper unfortunately.

2. Unclear threat modeling: does the adversary have access to the model training? If the adversary has access to the model training, which I believe is the case in the paper's setup, this type of assumption as backdoors is too strong in most practical setttings.


3. The formulation of catastrophe detection game in the paper does not have sufficient formalization. The problem are stated in the mathetical format but lacks meaningful substance. There are several statements mainly from intuitions without clear reasoning, including the following which does not have any formalized relationship in the definition:
> An intuition for requiring no catastrophes on other inputs is that we model rare but severe failure modes where the model uses an environmental cue to decide to deploy the catastrophic outputs. Frequent and unconditional catastrophes might be detected by simpler methods.


### Minor issues:
* The introduction section is missing section title.
* Line 126: Figure numbering is missing.

**Questions:**

1. The reasoning in the following statement does not seem to make sense, could you elaborate:
> An intuition for picking this trigger randomly
is that the input cues which are suitable for launching a catastrophic outcome are influenced by the
environment

2. When you introduce a new way to construct models, why is it reasonable to ignore evaluating the performances of the models in a more quantitive way? The paper seems to be missing that too.

---

### Official Review · Reviewer_hWmN · 2025-10-30

**Soundness:** 3
**Presentation:** 1
**Contribution:** 1
**Rating:** 2
**Confidence:** 3

**Summary:**

This paper proposes a backdoor attack based on cryptographic circuits in transformers that evade runtime detection. It formalizes runtime catastrophe detection as a two-player game: an attacker embeds behavior that produces a “catastrophe” only on a trigger input, while a defender who having white box access to model is deciding whether catastrophe will be triggered. Based on standard cryptographic assumptions (LWE and the decision version of LWE), the authors construct solving catastrophe detection as solving DLWE, which is not feasible in polynomial time without knowing the trigger. To prove this, they implemented a framework Reifier to map algorithm circuits into MLP blocks, which is more parameter-efficient comparing to current methods.

The experiment on the complied modules compares a standard password-locked backdoor with the proposed backdoor,  showing that probing can detect the former but not the latter. The paper suggests these modules could be embedded into open-source LLMs.

**Strengths:**

- This work provides a parameter-efficient implementation on cryptographic circuits via MLP, which is compact, implantable, and feasible in engineering. Insights of LTC to MLP/SwiGLU is also valuable.

- Theory framing provides a unified upper-bound perspective towards defending backdoors during runtime monitoring.

**Weaknesses:**

The paper’s writing creates massive confusion about its **core research question** and the link between **theory and experiments**. It is very unclear to me of the core research question is — what is the paper proving? And what are the experiments trying to tell?

- Overall story: the intro claims a **runtime backdoor detection impossibility** under white-box access; section 2 then discusses the **hardness of distinguishing LWE solutions from uniform**. The experiments, however, evaluate a compiled **PRNG $h(\cdot)$** (implemented as reduced-round SHA3-224) with a one-layer probe at layer 19. The catastrophe itself is not instantiated in the experiment, nor are end-to-end backdoors in the language model measured.
- Proof:
    1. The text conflates (1) indistinguishability of final outputs and (2) indistinguishability of intermediate activations. (1) Without framing target output $f(x^*)$ explicitly as $(A, z)$, it is already “not distinguishable” to pull out target B from random-looking strings in [1]. For (2), it seems to come from the intermediate PRNG circuit layers and not from DLWE, which is not formally proved in the paper.

    2. “White-box” is ambiguous: which signals (weights, activations, gradients) are permitted? Even if a logical circuit is “unprobeable,” does that imply the entire Transformer is also unprobeable?

- Experiments
    - Figure 2
        - Except for using MLP layers vs. attention layers, it is unclear whether this design is meaningfully harder to probe than the SHA-256 locker implemented in [1].
        - Key experimental details are missing: train/test set sizes and sources, seeds/variance, MLP probe hyperparameters, password-locked backdoor implementation details, etc.
        - Although this is an empirical test (and experiments cannot prove impossibility), there is no sweep over layers and no alternative probes (e.g., logit-lens, energy checks) at all.
    - The paper argues the compiled MLP module (19 layers, ~1.25B params) could be embedded into an LLM such as Llama-3 70B after pruning or distillation, but provides no end-to-end evaluation to support that the attack can succeed and normal performance is not degraded. From an engineering perspective, whether the module can be embedded also depends on width/shape compatibility, depth/placement, and potentially reprogramming attention routing (which should also comply with the white-box indistinguishability assumption).

[1] Draguns, Andis, et al. "Unelicitable Backdoors via Cryptographic Transformer Circuits." Advances in Neural Information Processing Systems 37 (2024): 53684-53709.

**Questions:**

- Please address the concerns elaborated in the weakness part. Especially, please clarify the difference between this work and [1], whether the unprobeability is a novel security property based on this paper's method.
- Some writing/presenting issues
  - Fix refs (line 047 see Fig.?; line 126 see Figure ?); unify “Fig.” vs “Figure”.
  - Writing consistency (line 306 what is the relationship between Fig. 3 and “simplified regime achieves strong resistance”)

---

### Official Review · Reviewer_gZNu · 2025-11-01

**Soundness:** 3
**Presentation:** 2
**Contribution:** 2
**Rating:** 4
**Confidence:** 4

**Summary:**

This paper introduces "Unprobeable Backdoors," a novel attack designed to evade runtime detection in Transformer models. The authors construct a backdoor based on cryptographic circuits and formalize their approach within a "catastrophe detection game." The central claim is that, under standard cryptographic assumptions (namely the DLWE problem), no polynomial-time defender with full white-box access to weights and activations can detect the backdoor's activation with accuracy better than random guessing. The paper also presents "Reifier," a framework for synthesizing algorithms into the MLP layers of a Transformer, and provides empirical results demonstrating that this backdoor construction successfully evades standard probing methods.

**Strengths:**

1.  The "catastrophe detection game" is a novel contribution, offering a formal adversarial framework to analyze the theoretical limits of runtime detection.
2.  The "Reifier" framework is an interesting technical contribution in its own right. It demonstrates a method for efficiently embedding complex algorithmic logic (like cryptographic hashes) into MLP layers, claiming significant parameter efficiency over prior work.
3.  The work provides a clear "worst-case" theoretical upper bound on the capabilities of runtime detection in AI safety.

**Weaknesses:**

1. The definition of "catastrophe" as the DLWE problem is highly theoretical and far removed from practical backdoor tasks. While this is an interesting question, it makes me question the practical novelty of the paper for the claimed application. The authors should at least provide an empirical validation on a non-cryptographic, sufficiently complex, and realistic security use-case that leverages these properties.
2. The assumption that models contain cryptographically-grade circuits is a very strong one. Real-world models are unlikely to have such structures, meaning they might expose partial risks that make them detectable. The paper needs to better justify the feasibility of constructing such circuits in a practical scenario; under current assumptions, this seems highly unlikely.
3. The attack model assumes the trigger is selected by the environment. However, in almost all practical scenarios, the trigger is carefully designed by the attacker.
4. There is a lack of experiments involving the injection of this backdoor into a real model. This is only claimed, not supported by theory or empirical evidence.
5. The paper's conclusions are overstated. I am concerned that this work does not offer a significant contribution to the current field of backdoor research and that its strong claims could be misleading to the community.

**Questions:**

1.  Can the authors provide a more concrete, non-cryptographic example of a "catastrophe" that is similarly "easy to construct but hard to check" (like the code vulnerability example mentioned in the paper)? How would the paper's theory apply to such a realistic scenario?
2.  The threat model assumes the attacker can implant a cryptographically-grade circuit. How plausible is this precise circuit implantation in a real-world training or supply chain attack? If the circuit is *not* cryptographically secure, does the "unprobeable" guarantee still hold?
3.  The game setup assumes the trigger $x^*$ is drawn randomly from the distribution $D$. As the paper acknowledges in the limitations, a real attacker would *choose* a trigger (e.g., a specific, rare string). Do the theoretical results (i.e., defender indistinguishability) still hold if the attacker is allowed to *choose* an (potentially out-of-distribution) trigger?
4.  The authors claim the circuit could be inserted into a model like Llama 3 70B, but the experiments only appear to test the compiled circuit in isolation. Why was this injection experiment not performed? Validating the attack's effectiveness and stealthiness on an actual large-scale model would significantly strengthen the paper's claims.

---

### Official Review · Reviewer_F4MN · 2025-11-01

**Soundness:** 3
**Presentation:** 2
**Contribution:** 2
**Rating:** 4
**Confidence:** 4

**Summary:**

This paper investigates backdoor attacks in machine learning models that are designed to evade runtime detection. The authors propose an "unprobeable backdoor" based on cryptographic circuits and analyze its security through a formal "catastrophe detection game." This game models a defender with full white-box access. The paper proves that, based on the hardness of the Decision LWE (DLWE) problem, no polynomial-time defender can reliably detect the backdoor's activation. To experimentally construct this backdoor, the authors developed the "Reifier" tool, which compiles algorithms into the MLP layers of a Transformer. The experimental results (Fig. 2) show that a standard MLP probe fails to detect this backdoor, whereas it successfully detects a simpler "password-locked" backdoor.

**Strengths:**

1.  **Formal Security Proof:** A core contribution of the paper is its formal proof of the infeasibility of runtime detection under standard cryptographic assumptions. This theoretical result is clear and powerful.
2.  **Clear Attack Construction:** The attack mechanism (Algorithm 1), using a PRNG and XOR operations, is clean, elegant, and provides a clear intuition for why its output is indistinguishable between the trigger and non-trigger cases.
3.  **Strong Empirical Baseline:** The experimental comparison in Figure 2 between the "unprobeable backdoor" and the "password-locked backdoor" is compelling. It provides strong empirical validation for the theoretical construction, showing that the cryptographically-based design does indeed resist activation-based probing.

**Weaknesses:**

* It is unclear how this theory applies to detection methods that are not based on a single-step decision. Many backdoor detection methods rely on multiple interactions with the model and draw conclusions from the interaction data. While these methods may have practical limitations (e.g., performance overhead from many queries), they are viable and must be considered.
* The number of cryptographic rounds appears to have been reduced in the experiments.
* The paper only states that a solution is "almost guaranteed" to not exist (for a random sample) but does not provide a failure probability bound.
* The paper briefly mentions that white-box security is not equivalent to a black-box assumption but does not elaborate on this point.
* The paper is missing an error analysis, especially concerning the compilation and injection of the circuits.

**Questions:**

1.  **Regarding the detection model:** The "catastrophe detection game" appears to be *one-shot*: the defender observes the behavior on a single input $x'$ and makes a decision. However, many real-world detection methods might be *interactive*, where the defender can query the model multiple times to gather information. Does the paper's theoretical guarantee extend to a defender who is allowed (a bounded number of) interactive queries?
2.  **Regarding cryptographic strength:** For "computational tractability," the experiment reduces the Keccak rounds of SHA3-224 from 24 to 3. While the authors claim this "achieves strong probe resistance," does this not fundamentally weaken the "cryptographic" indistinguishability guarantee the backdoor relies on? Is it possible the probe's failure is simply due to this simplified 3-round hash still being sufficiently obfuscating, rather than true cryptographic hardness?
3.  **On probability bounds and error:**
    * The paper states it's "almost" guaranteed that a random $(A', z')$ has no solution. Can a more precise failure probability bound be provided for this "almost"?
    * Given that the "Reifier" framework compiles algorithms into neural networks using weight ternarization and (presumably) floating-point arithmetic, has an analysis of the numerical stability and approximation error been conducted? How robust are these circuits to inference-time precision errors (e.g., in bfloat16)?

---

### Note · Authors · 2026-01-22

**Comment:**

We thank the reviewers for their time and feedback. We are withdrawing to submit a revised version to another venue.

**Withdrawal Confirmation:**

I have read and agree with the venue's withdrawal policy on behalf of myself and my co-authors.